# Ammonia Emissions from Cattle Manure under Variable Moisture Exchange between the Manure and the Environment

Rolandas Bleizgys and Vilma Naujokienė *

Department of Mechanics, Energy and Biotechnology Engineering, Vytautas Magnus University,
LT-53362 Kaunas, Lithuania; rolandas.bleizgys@vdu.lt
* Correspondence: vilma.naujokiene@vdu.lt; Tel.: +370-867358114

**Abstract:** When reducing ammonia emissions from cowsheds, it is recommended to reduce the ventilation intensity, air temperature in the barn, manure moisture by using bedding and manure-contaminated surfaces, and to prevent urine from accumulating in the airways. Using the mass flow method in the wind tunnel, after research on seven types of cattle manure with different moisture contents, it was found that ammonia evaporates up to 3.9 times more intensively from liquid manure than from solid manure. There is a strong correlation between ammonia and water evaporation from manure. Ammonia emission from liquid manure decrease by 2.0–2.3 times, emissions from solid manure decrease by 1.9–2.1 times. Different cowsheds have different opportunities to reduce air pollution and conditions for manure to dry and crusts to form on the surface. The best results will be achieved by applying complex measures to reduce air pollution.

**Keywords:** $NH_3$ emissions; manure; cowsheds; humidity; dry matter





## 1. Introduction

Ammonia is the main gas that acidifies precipitation, damaging the ecosystem. These two gases pollute the atmosphere and harm livestock kept in stables. Even low concentrations of $NH_3$ (10 mg m$^{-3}$) in the barn air worsen the health of the animals, damage the upper respiratory tract and the mucous membranes of the eyes, increase blood pressure, damage the heart, and reduce productivity and the animals' resistance to diseases. If the ammonia concentration is above 20 mg m$^{-3}$, immediate measures must be taken to reduce the concentration [1]. The permissible concentration of ammonia gas in stable air is regulated by individual states. In Lithuania, the ammonia concentration is limited to 15 ppm 11 mg m$^{-3}$ in the rooms where calves are raised and to 25 ppm 18 mg m$^{-3}$ in the stalls for young and adult cattle [1]. Ammonia consists more than 90% of total $NH_3$ in livestock and evaporates at all stages of manure formation, storage and spreading on the soil [2]. Ammonia emission processes in animal husbandry receive more attention than other gases [3]. Numerous studies have been performed analyzing microclimatic factors, as well as their importance and influence in cowsheds: air velocity [4,5], air temperature [6–9], relative humidity [10] and manure-contaminated surface areas [11]. Cattle are an important source of emissions and have the highest emission factors compared to other animals [12]. In the German livestock industry, most ammonia is emitted in naturally ventilated barns [13]. This view is shared by researchers in other countries [14]: naturally ventilated dairy barns are one of the main sources of ammonia and greenhouse gas emissions to the atmosphere. It is difficult to clean the air in these barns. To reduce emissions, the use of a hybrid ventilation system is recommended, in which the barns are naturally ventilated, and the air is exhausted from the slurry ducts with a mechanical ventilation system [15].

When modernizing cow-housing techniques, it is important to create a suitable production environment for productive cows. Increasing productivity was an important factor in moving away from tethered housing. In the future, more attention must be paid to barns

to ensure that cows behave naturally, that the climate is controlled, and that ammonia and greenhouse gas emissions are reduced [16].

Large-scale industrial farms are taking the place of livestock farms, which provide better opportunities to take measures to reduce $NH_3$ emissions. To reduce ammonia emissions, it is recommended to muck out the barn more frequently, separate urine from the thick liquid, and use rubber mats in the walkways [17]. Frequent cleaning of the barn floor and acidification of manure are effective means of reducing ammonia emissions [18]. Manure acidification technology has a significant impact on reducing NH3 concentrations in barns and can reduce $NH_3$ emissions by 45–60% [19]. The potential to reduce ammonia emissions by using urease inhibitors is up to 40% [13]. Ammonia emissions are strongly influenced by air temperature and relative humidity. Emissions increase with increasing temperature and decrease with increasing humidity. Microclimate factors have a greater influence on ammonia emissions than manure management system and soil and canal condition. As relative humidity increases, ammonia emissions decrease. This is due to the lower animal activity as less urine and feces are excreted [20]. Due to this large change, no significant effect of relative humidity on ammonia emissions was found [21]. Gas emissions vary greatly from day to day and year to year. When air temperature increases, ammonia emissions increase exponentially [22]. Other researchers have highlighted temperature as the most important factor influencing ammonia emissions [21]. A strong correlation between ambient air temperature and ammonia emissions has been found in the temperature range from 5 to 14 °C. As the ambient temperature rises, crust formation on the manure surface intensifies. Due to the crust formed on the manure surface, the intensity of ammonia emission from manure decreases [23]. In order to obtain reliable data on gaseous emissions from manure, the processes affecting the intensity of $NH_3$ emissions must be accurately described [24,25]. This makes it difficult to change the many factors that influence emissions. Ammonia emissions also change significantly with changes in air velocity, turbulence, and temperature [26,27]. Changes in ammonia emissions are also significantly affected by the crust on the surface of the slurry. The formation of this crust is highly dependent on the amount of straw, the dry matter of the slurry, and the ambient climatic conditions [28].

Ammonia evaporation is a diffusion process in which molecules of a gaseous substance move according to the concentration gradient from sites of higher concentration to sites of lower concentration until they are uniformly distributed. The volatilization of ammonia is described as a mass transfer process. When ammonia evaporates from the sludge, mass transfer occurs between the liquid at the surface of the sludge and the surrounding air stream. This evaporation process conforms to a common structure for all evaporation processes, and the basis of this structure is convective mass transfer, where the fluxes in ammonia vary as a function of the convective mass transfer coefficient and the $NH_3$ concentration gradient at the surface of the sludge layer and in the air stream above the sludge [29,30]:

$$E_{NH_3} = k_m(C_m - C_a), \tag{1}$$

Here, $E_{NH_3}$ is the ammonia evaporation flow density, mg m$^{-2}$ s$^{-1}$;

$k_m$—convective mass transfer coefficient, m s$^{-1}$;

$C_m$—gaseous ammonia concentration on the manure surface, mg m$^{-3}$;

$C_a$—concentration of ammonia in the ambient air stream above the manure, mg m$^{-3}$.

As the temperature of the manure increases, the emission of ammonia increases and a crust does not form on the surface of the manure because the high temperature of the manure increases the formation of aqueous ammonia, meaning that the dissociation constant and the concentration of protons in the solution increase. High manure temperatures increase the formation of gaseous ammonia and reduce the solubility of ammonia in water. The aim of the study: to determine the influence of manure moisture (the amount of dry matter in manure) on the evaporation of ammonia from cattle manure and the possibilities of reducing ammonia emissions from cowsheds by controlling the thermal and technological factors of manure handling.

## 2. Materials and Methods

### 2.1. Research Methodology in Laboratory Conditions

A trainer was designed and fabricated (Figure 1), in which the process of ammonia gas evaporation from manure and the effect of various factors on this process were analyzed by simulating the conditions in cowsheds. The values of the thermal factors that have the greatest influence on the evaporation of ammonia from the manure correspond to those in the stall. These are the velocity of air movement, temperature, and humidity. The velocity of air flow over the manure varies between 0 and 1 m s$^{-1}$, and the air temperature is 12–25 °C. To avoid air turbulence in the wind tunnel, its length is more than 10 times larger than its diameter. The cross-section of the wind tunnel is a square with sides 0.40 m long. The length of the wind tunnel is 6.0 m. The wind tunnel is made of chemically stable material: stainless steel.

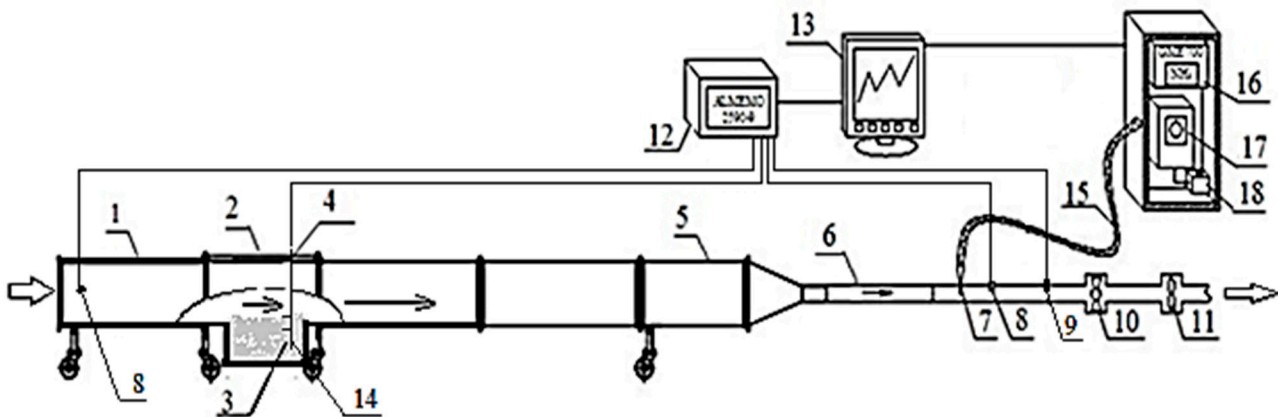

**Figure 1.** Scheme of ammonia emission from manure in a wind tunnel: 1—wind tunnel; 2—cover; 3—manure storage chamber; 4—section for storing manure; 5—transition cone; 6—duct (100 mm diameter); 7—air sampling probe; 8—temperature and humidity sensors; 9—thermoanemometer sensor; 10—the valve; 11—fan with frequency converter; 12—meter-accumulator "Almemo 2590-9"; 13—computer (AMR program); 14—thermopores; 15—heated air supply hose; 16—laser gas analyzer "GME700"; 17—electrically heated three-channel valves; 18—diaphragm air pump.

A mass flow method was used to investigate the emission intensity. Knowing the ventilation intensity $G$ (m$^3 \cdot$h$^{-1}$) of the chamber, the gas concentration Ce entering the chamber (mg m$^{-3}$) and the air $C_o$ leaving it, the gas emission intensity $E_{NH_3}$ is calculated:

$$E_{NH_3} = (C_o - C_e)\, G \qquad (2)$$

The tests were performed with fresh manure taken from the cowshed; the dry matter content was determined an gas emission tests were performed. The manure was drained and leveled in a 0.23–0.25 m thick layer in the manure storage chamber (3). The manure chamber was placed in a wind tunnel in the manure storage section (4). The fan (11) drew air from the wind tunnel to create the required air flow over the manure. The fan was installed in the air extraction duct (6) with a diameter of 100 mm. The length of the duct was 1.50 m, i.e., more than 10 times longer than diameter. This helps prevent air turbulence in the duct. The ventilation intensity of the wind tunnel was changed by changing the cross-sectional area of the duct in the air outlet duct (6) and by changing the speed of the fan (11) with a frequency converter. Air samples were taken from the duct (6) by probes (7) and the heated hose (15) was fed to the gas analyzer (16). The air was supplied to the analyzer continuously by a pump (18) with a capacity of 6 L min$^{-1}$. To prevent air condensation, it was heated in a hose (15) and heated to 150 °C in electrically heated valves (17). At the start of the test, the Almemo 2590-9 gas analyzer (AHLBORN Thuringia, Germany) and data logger (12) were compatible with each other and programmed to record data every 1 min.

Ammonia (NH$_3$) gas concentration is measured with a GME700 analyzer (SICK MAI-HAK GmbH, Reute, Germany) (16). The measurement ranges from 0 ppm to 2000 ppm. The measurement principle of this instrument is laser spectroscopy, and the accuracy is 2–4%. The response time depends on the air flow and gas concentration and is less than 360 s. At an air flow of 6 L min$^{-1}$ and an ammonia concentration in the air of 30 ppm or less, the analyzer accurately measures the ammonia concentration in the air within 60–80 s, and the readings stabilize during this time. The analyzer is equipped with a heater for the pumped gas, so the gas conditioning ensures that the cell is not contaminated, and condensate does not form in the cell. Electrically heated three-way valves ensure that the operating temperature is higher than the dew point of the sample. Operating mode: automatic (continuous or cyclic measurement with data acquisition).

The temperature and humidity of the air entering and leaving the wind tunnel are measured by temperature and humidity sensors (8) connected to the Almemo 3290-8 system (Ahlborn GmbH, Germany) (12). Measuring range: temperatures from 30 to 60 °C; relative humidity from 5 to 98%; accuracy of the device ±0.1%. The temperature of the manure is measured by thermocouples (14). The thermocouple consists of moisture-resistant insulated wires with a diameter of 0.5 mm. Resolution of temperature measurement: 0.1 °C. The thermocouples are connected to the meter-accumulator "Almemo 2590-9" (Ahlborn GmbH, Ilmenau, Germany).

The temperature of the manure was measured with 9 thermal sensors (4) made of Cu–CuNi (copper–constant) of 0.5 mm in diameter. The minimum temperature measurement range was from −25 °C to 200 °C. Temperature measurement resolution—0.1 °C. Temperature measurements were recorded on a computer-controlled device Almemo 2590-9 (Ahlborn GmbH, Ilmenau, Germany).

The temperature and humidity of the air entering and leaving the manure chamber were measured with temperature and humidity sensors (8) connected to the Almemo 2590-9 system (14). Measuring range: temperatures from −30 to 60 °C; relative humidity from 5 to 98%; accuracy of the device ±0.1%. Almemo 2590-9 system is equipped with an anemometer (9), which measures the speed of air movement in the duct and converts it into air-flow rate. Measuring range 0–10 m s$^{-1}$, accuracy ±0.1 m s$^{-1}$.

The amount of dry matter in manure is determined based on the requirements of the standard (LST 1530:2004 Legumes, buckwheat and their products). Approximately 200 g of manure was added to each drying tank and an 8 cm high layer was formed. The sample prepared for drying was placed in a drying oven Memmert model 100–800 and dried at a temperature of 105 °C. The drying of the manure is finished when the manure is dried to a constant mass, and the manure samples are weighed. Based on the obtained weight, the amount of dry matter in the manure is calculated. At least eight repetitions were performed each time to determine the amount of manure dry matter.

The value of the hydrogen ion indicator (pH) in the manure was determined by the HI98129–HI98130 pH meter. A pH meter is an electronic device with measurement limits ranging from 0 to 14.0 pH, measurement accuracy—0.05 pH measured values.

The obtained experimental results were converted to conditional values: ammonia emission intensity per square meter of manure surface area (mg m$^{-2}$ h$^{-1}$); ventilation intensity per square meter of manure surface area (m$^3$ m$^{-2}$ h$^{-1}$).

### 2.2. Research Methodology in Production Cowsheds

For experimental research in production conditions, we chose three new development trends that are most suitable for dairy cowsheds: semi-deep insulated, cold box-type (box-type uninsulated), semi-warm box-type (semi-insulated box-type) (Figure 2).

| | |
|---|---|
| Semi-deep dairy cowshed—old stringed cowsheds are often reconstructed into such barns; they are popular on small farms. There are 140 cows in the cowshed. One cow has 8.6 m² of bedding area; the animal needs 9–11 kg of straw per day. Manure is removed from the bed by mobile equipment 1–2 times a month. The barn is equipped with a shaft ventilation system; the roof is insulated with a thick layer of straw. The width of the cowshed is 18 m; the length is 105 m. |  |
| Cold box dairy cowshed—suitable for medium-sized and large farms. There are 200 cows in the cowshed, which are kept in shallow stalls. Manure is removed from the path 8 times a day with a scraper conveyor. The walls and roof of the building are not insulated; the average heat transfer coefficient is 4.5 W·(m²·K)⁻¹. The barn is equipped with a ridge-slit ventilation system. Air circulation is controlled by changing the area of the slits in the walls. The width of the cowshed is 21 m; the length is 72 m. |  |
| Half-heated box dairy cowshed—such cowsheds are the most common built in Lithuania today, they are popular on medium-sized and large farms and suitable for the robotic milking of cows. A total of 250 cows are kept in the cow shed. The average heat transfer coefficient of the barn walls is 3.3 W·(m²·K)⁻¹, and that of the roof is 0.45 W·(m²·K)⁻¹. The roof of the barn is insulated. It uses litter-free technology; the boxes are lined with rubber mats, and the walkways are equipped with grills. Liquid manure is removed from the canals daily. The barn is equipped with a natural, slotted ventilation system. The width of the cowshed is 32 m; the length is 83 m. |  |

**Figure 2.** M1...M5 microclimate factor measurement location in the cowshed.

To determine the changes in factors affecting the moisture content of manure and crust formation on the manure surface, studies were conducted in barns at different times of the year. The changes in air temperature, relative humidity, velocity, and ammonia gas concentration in the barns and outdoors were determined. Air temperature and relative

humidity are recorded hourly with MicroLite LITE5032P- RH Fourtec (St Neots, United Kingdom) programmable meters. A total of 7 m was used for the research, 2 in the field and the other 5 in the barn above the beds. Temperature measurements range from $-35\,^\circ$C to $+85\,^\circ$C; measurement accuracy $\pm0.5\,^\circ$C readings. Relative humidity measurement ranges from 0% to 100%; measurement accuracy $\pm3\%$ readings.

To determine whether the microclimate meets animal welfare requirements and to evaluate the ammonia gas concentration in the barns, air samples were taken and the concentration was measured in the laboratory with a laser gas meter GME700 (SICK MAIHAK GmbH, Reute, Germany). Samples were taken from all places and brought to the laboratory under the same conditions and in the same special bags from which the air cannot escape. Air samples were taken in the places where temperature and air humidity were measured (at a height of 1.5 m above the floor) with a special device Ecoma CSD 30 (in 10 L bags). Taking one air sample takes 25 min. The air sampling device was made of odor-neutral materials: stainless steel and PTFE. The cylinder was made of transparent PVC. The bags are made of Nalophan. The materials of the air sampling equipment complied with standard (EN 13725:2003 Air Quality—Determination of odour concentration by dynamic olfactometr). The concentration of ammonia in the air sampled in the bags was measured by sucking air from them into the gas analyzer GME700. Air suction lasts up to 100 s and this amount of air (10 L) is enough to accurately measure the ammonia concentration. At low ammonia concentrations (up to 30 ppm), the gas analyzer records a stable concentration value within 60–80 s.

The speed of air movement in the cowshed was measured with a vane anemometer FVA915MA1 (Ahlborn GmbH, Ilmenau, Germany). Wing pipe diameter 80 mm; measurement limits 0.2–20 m/s; measurement accuracy $\leq\pm1.5\%$ measured values. The device, which is equipped with a 200 mm diameter air flow formation and intensification conical hood, is connected to the Almemo 2590-9 measuring and data collection device.

### 2.3. Statistical Evaluation of Data

The studies lasted from 164 to 228 days in cowsheds. The obtained data were analyzed by the polynomial correlation and regression method. The tests were repeated 6–8 times in the laboratory bench. Arithmetic means of indicators and their standard errors were calculated by analyzing the research data. The standard error and the minimum confidence interval were calculated at the level of statistical significance $p < 0.05$.

## 3. Results

### 3.1. Intensity of Ammonia Emissions, Water Evaporation Dependences on Differences between Manure Layer and Surface Temperatures

Controlling the evaporation of ammonia from manure is essential to reduce ammonia emissions. It is necessary to properly regulate the processes that take place on the surface of the manure, on which the moisture of the manure surface depends—the formation of the crust and the entry of oxygen to the manure. Ammonia evaporates more intensively from liquid manure. In the wind tunnel, emissions of ammonia ($p \leq 0.05$) from liquid cattle manure (dry matter 7.4%) were determined to be $404 \pm 13$ mg m$^{-2}$ h$^{-1}$, semi-liquid (dry matter 14.6%)$-223 \pm 11$ mg m$^{-2}$ h$^{-1}$, from the thick (21.9% of dry matter)—$103 \pm 7$ mg m$^{-2}$ h$^{-1}$ (Figure 3). During the tests, the temperature of the clean air supplied to the chamber was constant at 19.6–0.6 $^\circ$C, the relative humidity was $51.7 \pm 4.9\%$, and the chamber was ventilated at a constant intensity (4.03 m$^3$ h$^{-1}$ or 23.7 m$^3$ h$^{-1}$ per square meter of manure surface area). The average manure temperature during the tests was $19.8 \pm 0.3\,^\circ$C. The manure used for the research contained 0.36% of total nitrogen (N), 0.05% of ammonia nitrogen (N-NH$_4$); during the research, the average manure pH was $6.92 \pm 0.17$ (Figure 3).

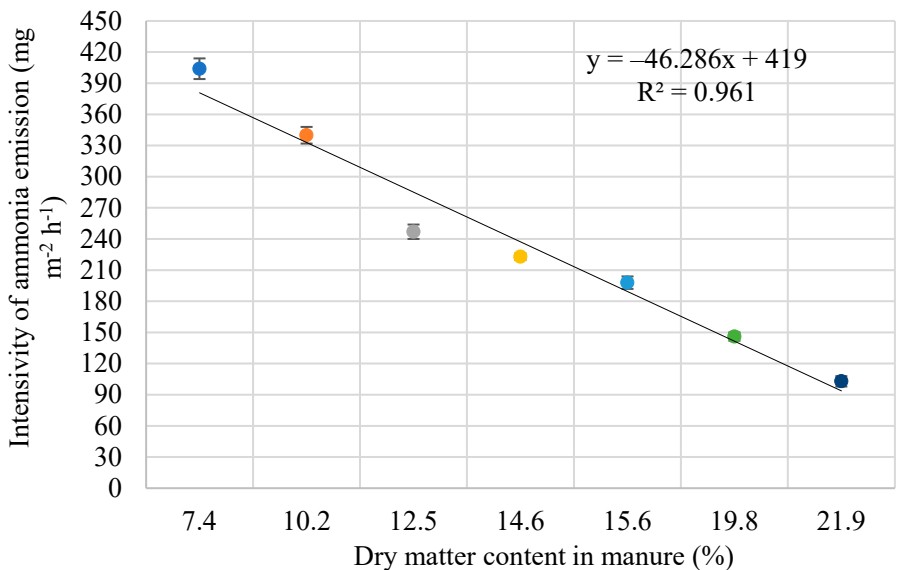

**Figure 3.** Intensity of ammonia emissions dependence on dry matter content in manure.

Significant differences were found between all values, so none of the columns were marked with a letter or other marking indicating that there were no significant differences between the values. The confidence intervals were calculated individually for each parameter according to the statistical evaluation methodology. Confidence intervals indicate the spread of replicate values and the scatter about the mean value. As a very large volume of data was evaluated, a regression analysis was performed using the mean values of the primary replicates. It has been established that as the values of one parameter increase the values of the other parameters change in a dependent manner.

When manure is stored, its surface dries and a crust forms on the surface, and ammonia emissions decrease. Due to the drying of the manure, the largest changes in ammonia gas emissions are from liquid manure. From this, ammonia emissions are reduced from 404 to 170 mg h$^{-1}$ m$^{-2}$ in 24 h, i.e., emissions are reduced by almost 60% (Figure 4). Ammonia emissions from slurry are reduced from 103 to 49 mg h$^{-1}$ m$^{-2}$.

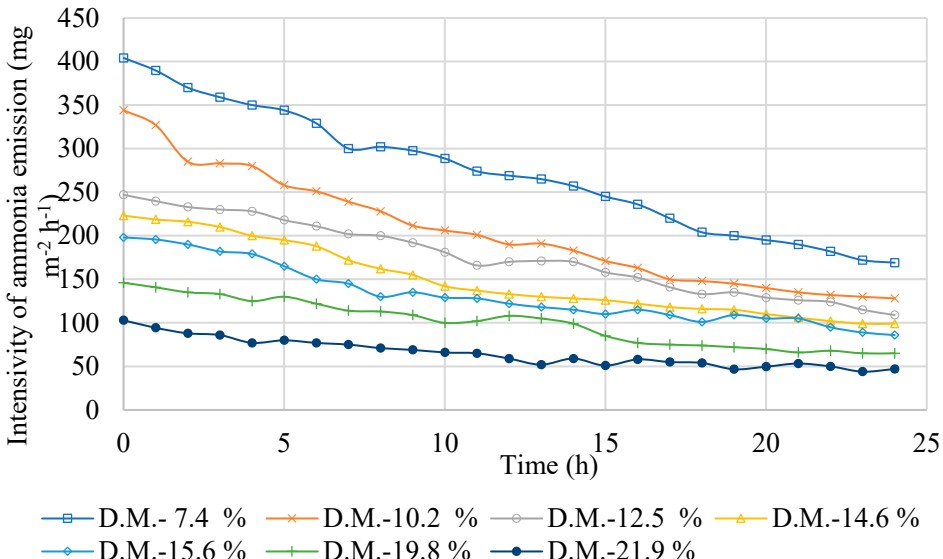

**Figure 4.** Intensity of ammonia emissions from cattle manure of varying humidity stored in the wind tunnel for 24 h.

The evaporation of ammonia from manure is related to the intensity of moisture evaporation. When the manure is stored, water evaporates intensively (Figure 5), the moisture of the manure surface decreases and a crust is formed, which reduces ammonia emissions. In 50 h, the evaporation rate of water from fresh liquid manure is reduced by about two-fold (from 150 g m$^{-2}$ h$^{-1}$ to 80 g m$^{-2}$ h$^{-1}$) and ammonia emissions are reduced by more than three-fold (from 340 mg m$^{-2}$ h$^{-1}$ to 100 mg m$^{-2}$ h$^{-1}$). During the tests, the temperature of the clean air supplied to the chamber was constant at 20.3 ± 0.8 °C, the relative air humidity was equal to 49.4 ± 3.7%, and the chamber was ventilated at a constant intensity. During the research, the average manure temperature was 20.7 ± 0.29 °C. The manure used for the research contained 0.34% of total nitrogen (N) and 0.05% of ammonia nitrogen (N-NH$_4$), and the average pH of the manure during the research was 7.03 ± 0.21.

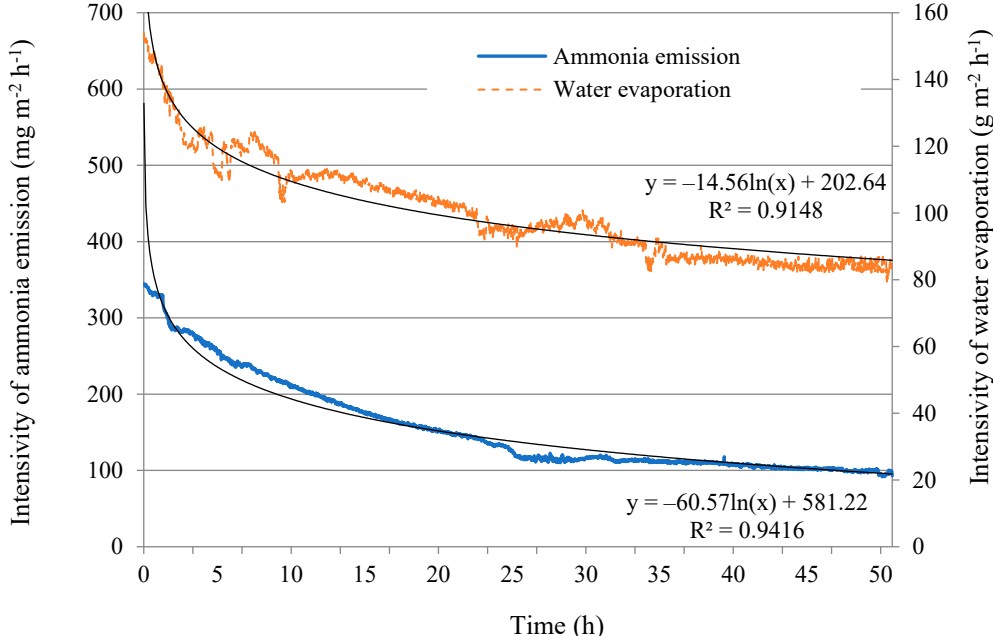

**Figure 5.** Intensity of ammonia emission dependence on intensity of water evaporation from manure (manure D.M.-10.9 ± 0.38%).

Although the evaporation intensity of ammonia and water from manure varies, a strong correlation was found between these values (Figure 6), according to which the intensity of ammonia emission can be predicted by the evaporation intensity of water from manure. A strong correlation between ammonia emissions and water evaporation from manure was also found in studies with manure with different moisture contents: D.M.-7.1 ± 0.29%; D.M.-10.9 ± 0.38%; D.M.-13.2 ± 0.36%; D.M.-22.7 ± 0.51%.

During the 50 h study, the dry matter content of the surface manure layer increased to 36.3 ± 5.4%. Accurately determining the moisture of the surface layer (crust) is a methodological problem involving taking manure samples from the surface. The change in moisture evaporation intensity from the manure surface and its drying can also be inferred from the temperature change in the layer vertical (Figure 7). As the intensity of ammonia and, thus, water evaporation slows down, the temperature difference between the manure layer and the surface decreases.

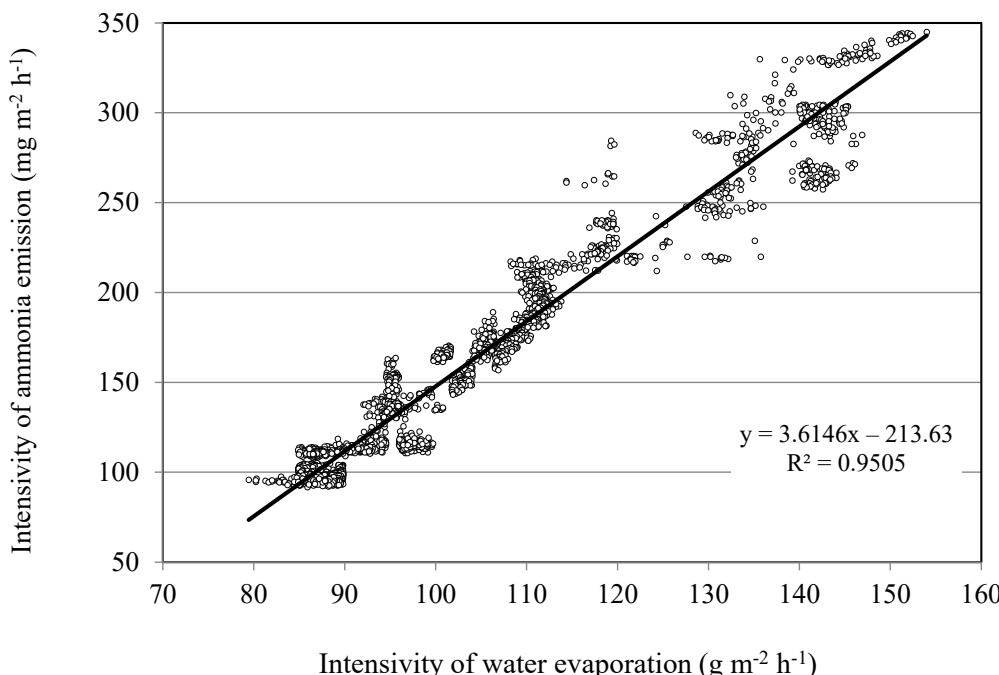

**Figure 6.** Intensity of ammonia emission dependence on intensity of water evaporation from manure (manure D.M.-10.9 ± 0.38%).

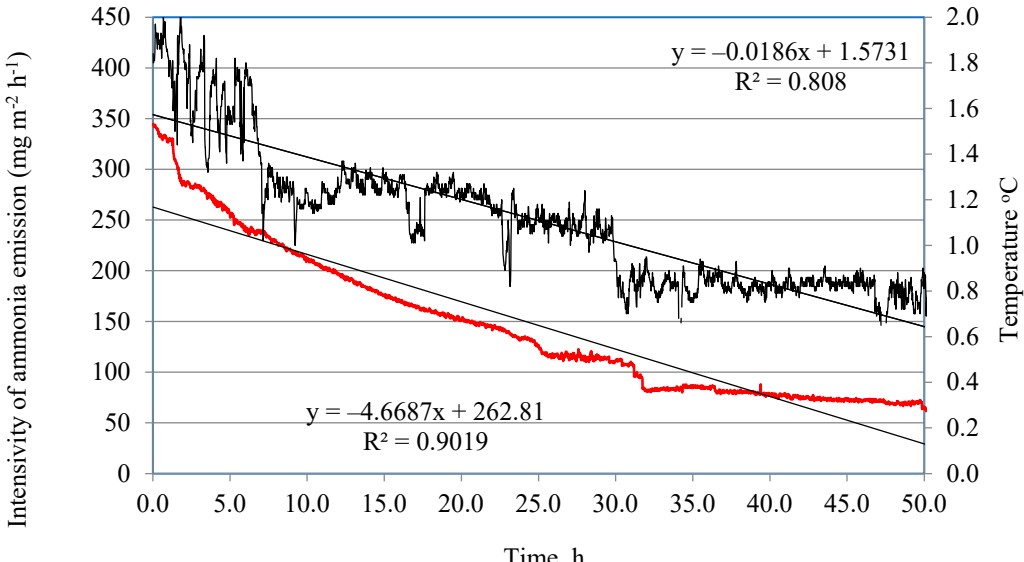

**Figure 7.** Change in ammonia emissions, manure layer and surface temperature difference (manure D.M.-10.9 ± 0.38%).

As the surface of the manure dries, the evaporation intensity of water and ammonia from the surface decreases, and the temperature difference between the manure layer and the surface decreases (Figure 8). A crust forms on the surface of the manure, which reduces ammonia emissions.

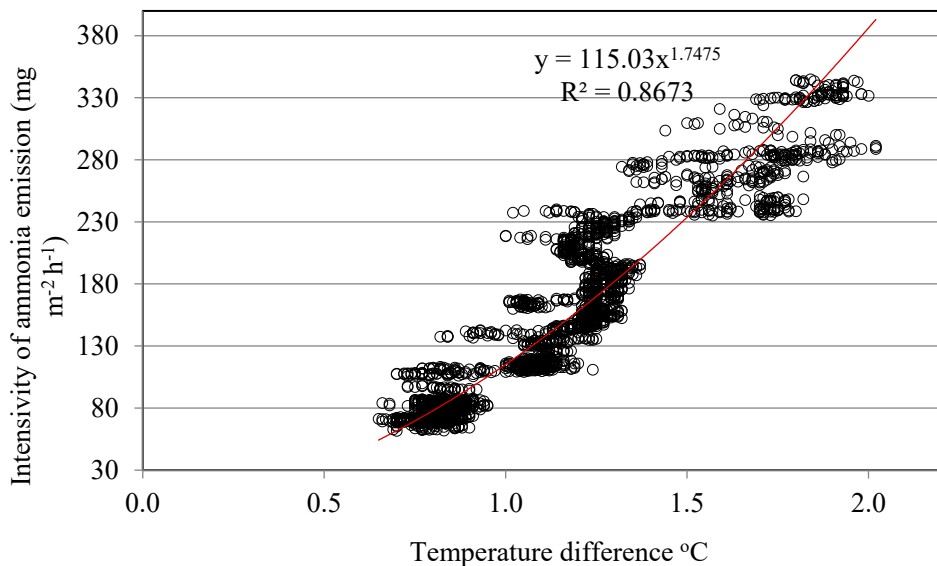

**Figure 8.** Dependence of ammonia emission intensity on the difference between manure layer and surface temperatures (manure D.M.-10.9 $\pm$ 0.38%).

Evaporating water from manure increases the concentration of aqueous ammonia. If the concentration of ammonia in the air is lower than the concentration above the manure surface, intensive ammonia evaporation takes place. The humidity of the manure surface, as well as the evaporation of ammonia, are affected by the ambient temperature, air humidity and the effect of solar radiation. As the manure dries, a crust forms on its outer layer, which acts as a protection against ammonia diffusion.

### 3.2. Evaluation of Thermal Factors of Ammonia Emission in Cowsheds

In order to reduce ammonia emissions from cowsheds, it is very important to prevent liquid manure from accumulating in the tracks, to increase the dry matter content of manure, and to allow for the formation of crusts on the surface. Manure drying and crust formation are highly dependent on microclimate factors: air temperature, humidity, speed of movement. Factors influencing the ammonia emission process also change with changes in air temperature, ventilation intensity (air flow over manure): manure temperature, ammonia concentration gradient above manure, manure surface moisture, and conditions for crust formation.

The cowsheds in which the research was carried out use various technologies and create very different conditions for the formation of crusts on the surface of manure. With the growth of farms and the reduction of labor costs, less use is made of bedding (3rd cowshed). Litter-free technologies are becoming more popular, and more liquid manure is being stored. The barns are equipped with manure disposal channels, which are covered with gratings. In order to reduce ammonia emissions, it is important to allow for manure to dry and form crusts on the surface. There are better conditions for manure drying in smaller farms where thick manure is stored (1st cowshed).

After conducting research in the most popular cowsheds in Lithuania, very different conditions for manure drying in cowsheds were determined. Air speed in a cold cowshed is 0.12–0.18 m s$^{-1}$ in winter, 0.15–0.21 m s$^{-1}$ in a warm one, and in summer the air speed in a cowshed increases to 1.60 m s$^{-1}$. In a cold cowshed, air speed is 1, 38–1.59 m s$^{-1}$; in semi-warm weather, it is 1.31–1.54 m s$^{-1}$. The temperature varies from 3.9 °C to 27.1 °C in the semi-deep cowshed and from −0.8 °C to 29.1 °C in the semi-warm cowshed. In cold cows, the temperature changes the most: from −14.1 °C to 33.2 °C. In insulated barns, the temperature changes less: during the heat, it is only a few degrees lower than in the cold cowshed, and during the cold it does not fall below 0 °C. During the heat, the temperature is too high in all barns; in uninsulated cows it is 5–8 °C higher than the

recommended maximum (25 °C), and in insulated cows it is higher than 4–5 °C. At these temperatures, the evaporation of ammonia from the manure intensifies greatly. Different temperatures in the barns are determined by the construction of the barn, the insulation of the building enclosures, the stocking density and the intensity of ventilation. Comparing the air temperatures in the cowsheds with the insulated roof with the cold cowsheds, significant temperature differences were only found in the barns during the cold, and the temperature did not differ much during the heat.

In cowsheds, the main problem is not bad temperatures or high ammonia concentrations, but high humidity. When the outdoor temperature drops, a positive temperature in the barn is achieved by reducing the intensity of ventilation and deteriorating the cleanliness of the air. When the outdoor temperature drops below 0 °C, the temperature difference between the barn and the outside increases to 15.0 °C, and when it is very cold outside (−23.0 °C), the temperature difference increases to 26.0 °C. This indicates insufficient ventilation intensity. As a result, the relative humidity of the air is close to 100%. Therefore, there are no possibilities of increasing the relative humidity of the air in cowsheds in order to reduce the evaporation of ammonia from manure.

No correlation was found between ammonia concentration and relative humidity in the cowshed (Figure 9). This was due to a large change in the factors influencing the evaporation of gas from the manure: air temperature and ventilation intensity.

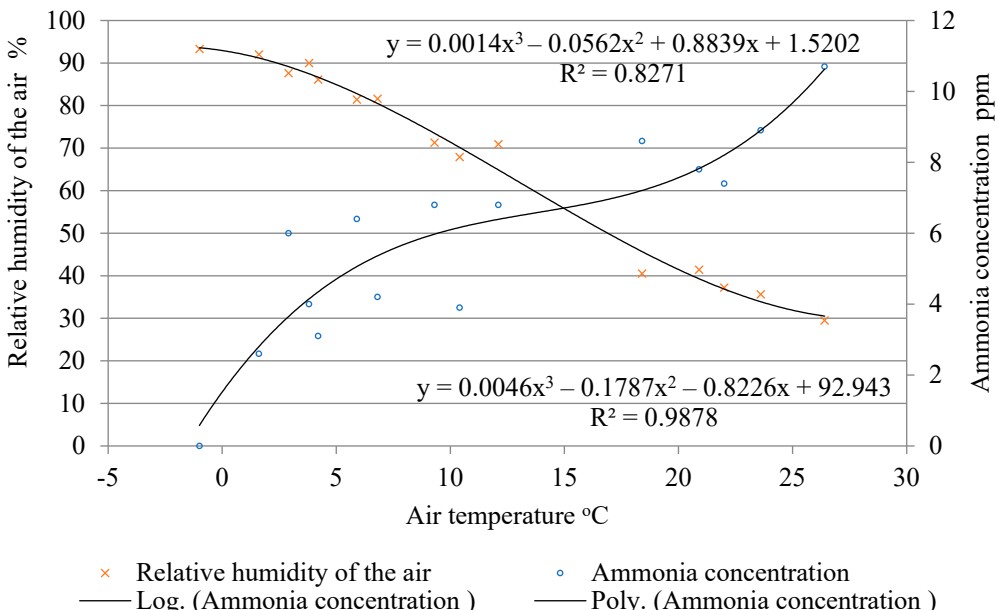

**Figure 9.** Ammonia concentration dependence on relative humidity of air in a semi-warm cowshed on air temperature.

At low temperatures and high humidity in the barn, ammonia concentration is not always low (Figure 10). At that time the ventilation channels are very closed and the intensity of ventilation is greatly reduced, which affects the increase in ammonia concentration. When the temperature increases and the air humidity decreases in the barn, the concentration of ammonia does not always increase proportionally, because when it is warm outside, the ventilation ducts are opened and the barns are ventilated very intensively. For example, in a semi-warm cow house at a temperature of 2.5–5.9 °C, air speed varied from 0.31 to 0.45 m s$^{-1}$, the ammonia concentration varied within the following limits: 3.1 ± 0.06–6.4 ± 0.09 ppm; with an increasing temperature above 20.0–26.4 °C, air speed increased to 0.97–1.32 m s$^{-1}$, and the range of ammonia concentration change remained similar: 7.8 ± 0.08–10.6 ± 0.11 ppm. Even in the event of a drop in temperature, a decrease in ventilation intensity can lead to a poor microclimate.

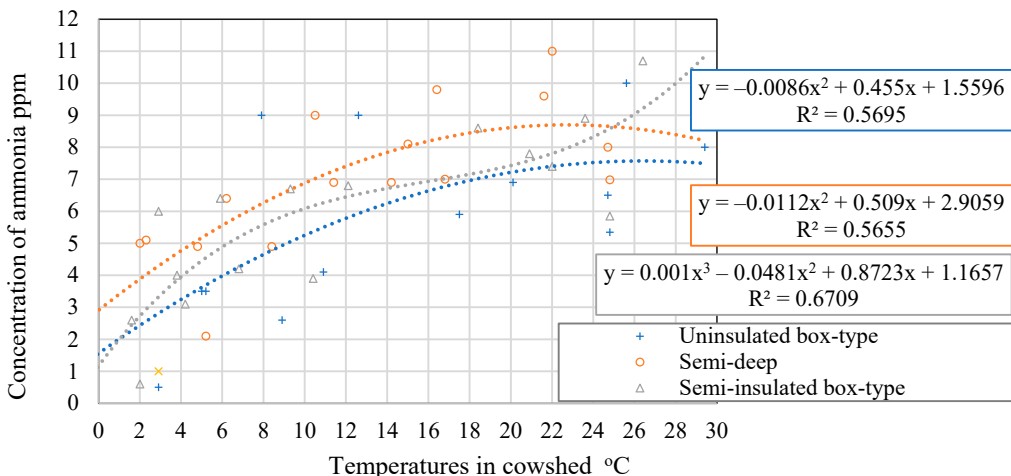

**Figure 10.** Ammonia concentration dependence on temperatures in cowshed.

It is recommended to reduce air pollution with ammonia by applying complex measures: controlling thermal processes in cowsheds and optimizing manure management (Table 1). The application of different measures depends on the cowshed and the technology used.

**Table 1.** Reduction in ammonia emissions from cowsheds.

| Ammonia Emissions Mitigation Measure | Semi-Deep Insulated Cowshed | Box-Type Uninsulated Cowshed | Box-Type Semi-Insulated Cowshed |
|---|---|---|---|
| Reduce the surface area of manure by more intensively removing manure from paths into canals | Low opportunities | Good opportunities | Good opportunities |
| Reduce manure-contaminated surfaces | Low opportunities | Good opportunities, but limited by animal welfare conditions | Good opportunities, but limited by animal welfare conditions |
| Reduce manure moisture by using bedding | Good opportunities | Medium opportunities | Low possibilities, popularity of sledgehammer technology |
| Do not allow urine to accumulate in the airways | Medium opportunities | Good opportunities | Good opportunities |
| Mix less manure | Low opportunities | Low opportunities | Low opportunities |
| To increase the humidity in the cowshed | Limits animal welfare conditions | Limits animal welfare conditions | Limits animal welfare conditions |
| Use biological manure treatment | Medium opportunities | Good opportunities | Good opportunities |

## 4. Discussion

Manure is liquid when it contains less than 12% DM; semi-liquid—12–20% DM; solid manure has more than 20% DM.

The highest ammonia emissions are from liquid manure. Ammonia evaporates intensely from the urine, breaking down the urea. Therefore, ammonia emissions from urine-coated manure are much higher than those from urine-free manure. Straw or other bedding absorbs moisture, reduces the amount of liquid on the surface of the manure and reduces the evaporation of ammonia. The highest ammonia emissions occur in cowsheds from surfaces contaminated with liquid manure and where urine accumulates, and the lowest occur above the thickened manure on which litter is applied [31].

Even in the event of a drop in temperature, a decrease in ventilation intensity can lead to a poor microclimate. This statement is confirmed by the research conducted by [32], according to which excessive reduction in ventilation intensity in the barn is the most common cause of deteriorations in indoor air quality [33].

Ammonia is released from manure as a result of urease activity, which is influenced by the temperature of the manure, and the temperature of the manure is highly dependent on the air temperature. Ammonia concentration in naturally ventilated sheds is mainly influenced by air temperature. An increase in temperature during the summer increases urease activity, and thus increases ammonia emissions from manure [34]. However, even with very high ammonia emissions from manure, the concentration of these gases can be low, with a very high aeration.

The thermal factors of the barn have a significant effect on the concentration of ammonia in the barn air. For a more detailed analysis, data on the influence of temperature, humidity, and ventilation intensity on the ammonia emission process are required for the ammonia emission process and the concentration in the barn air. Determining the effect of these factors on emissions in the barn is difficult because all the factors vary over a very wide range, and in naturally ventilated barns it is difficult to determine the exact intensity of ventilation. These results are confirmed by studies by other researchers [30,35–37]. The values of ammonia concentrations are very different in different cowsheds, but their trends are similar in different periods of the year.

The concentration of ammonia in the barn air is a factor when assessing the cleanliness of the air. However, this cannot determine ammonia emissions into the atmosphere. The impact of ammonia on livestock, air quality in barns, and atmospheric pollution with these gases should be addressed in a comprehensive manner, assessing air quality in the animal housing (gas concentration and air humidity) and ammonia emissions to the atmosphere (emission intensity). These processes are mainly influenced by the intensity of ventilation, which affects the cleanliness of the air in the barn, as well as microclimate factors (air humidity), and has a significant impact on the emissions of gaseous pollutants into the atmosphere. Therefore, to ensure good air quality in the barn and to reduce atmospheric pollution with harmful gases, it is important to properly manage the ventilation intensity. As the ventilation is intensified, the ammonia evaporates more. Therefore, if the relative humidity and carbon dioxide concentration in the barn are within normal limits and the ammonia concentration is high, ways to reduce this should be sought, but not by intensifying ventilation [38].

During heat, it is recommended to cool the air in the barn or use biological manure treatment that reduces ammonia emissions by more than 30%. The maximum effect of the biological treatment of manure on ammonia emissions occurs after 28–35 days [39].

Nitrogen losses on the farm are related not only to proper manure management, but also to animal feeding. The composition of the animal diet affects the release of nitrogen and carbon into the environment. In cattle fed high-concentrate forages, the optimum crude protein concentration in the diet to reduce unwanted carbon and nitrogen losses is 14.5%. This is a good strategy to reduce $NH_3$ emissions [40]. The most important ruminant nutritional strategies to reduce N emissions found that reducing the crude protein concentration in the diet was one of the most effective methods by improving microbial urea utilization in the rumen [41]. For example, feeding black soldier fly larvae is envisioned as a novel circular strategy to extract manure, reduce its environmental impact, and convert it into insect protein for use in animal feed. Using the stable isotope 15N $NH_3$, at least 13% of pig manure $NH_3$-N is incorporated into fly larval body mass, increasing $NH_3$-N uptake into larval proteins, and thereby reducing $NH_3$ release from manure [42].

Ammonia ($NH_3$) volatilization from fertilization and animal waste is an extremely important mode of nitrogen loss in agricultural ecosystems and is the largest source of atmospheric $NH_3$. The volatilization of $NH_3$ is highly dependent on environmental and meteorological conditions [43].

## 5. Conclusions and Recommendations

Ammonia evaporation is related to the intensity of moisture evaporation from manure; a strong correlation between these parameters has been established. Ammonia evaporates up to 3.9 times more intensively from liquid manure than from solid manure. When manure

is stored, water evaporates intensively, the humidity of the manure surface decreases, and a crust forms, which reduces ammonia emission. In 50 h, the intensity of water evaporation from fresh liquid manure decreases by about two times; ammonia emissions decrease by more than three times. Due to the drying of manure and the formation of a crust on the surface, the biggest changes in ammonia emissions occur from liquid manure. Ammonia emissions decrease to 230 mg h$^{-1}$ m$^{-2}$ intensity over 24 h (2.0–2.3 times), while those from thick manure decrease to 54 mg h$^{-1}$ m$^{-2}$ intensity (1.9–2, 1 time).

Manure drying and crust formation are highly dependent on microclimate factors: air temperature, humidity, speed of movement. As air temperature and ventilation intensity (air-flow over manure) change, the factors influencing the ammonia emission process also change: manure temperature, ammonia concentration gradient above manure, manure surface moisture, and the conditions for crust formation.

The best results were achieved by applying complex measures to reduce air pollution in cowsheds: increasing the air humidity in the barn, without exceeding the recommended limits (in production cowsheds, it was established that, due to the high air humidity, the possibility of applying this measure is low); reducing the moisture content of manure by using bedding; not breaking up the crust formed on the surface of the manure; preventing urine from accumulating in the paths in the barn and on the manure surface; reducing surfaces contaminated with manure.

**Author Contributions:** Conceptualization, R.B. and V.N.; methodology, R.B. and V.N.; software, R.B. and V.N.; validation, R.B. and V.N.; formal analysis, R.B. and V.N.; investigation, R.B. and V.N.; resources, R.B. and V.N.; data curation, R.B. and V.N.; writing—original draft preparation, R.B. and V.N.; writing—review and editing, R.B. and V.N.; visualization, R.B. and V.N.; supervision, R.B. and V.N.; project administration, R.B. and V.N. All authors have read and agreed to the published version of the manuscript.

**Funding:** This research received no external funding.

**Institutional Review Board Statement:** Not applicable.

**Informed Consent Statement:** Not applicable.

**Data Availability Statement:** All data are available from the authors upon reasonable request.

**Conflicts of Interest:** The authors declare no conflict of interest.

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
