# Peer review of "Ammonia Emissions from Cattle Manure under Variable Moisture Exchange between the Manure and the Environment"

_agronomy, doi:10.3390/agronomy13061555_

Round 1
Reviewer 1 Report
This paper focuses on the evaluation of ammonia emissions with different levels of moisture content.
Line 39: I would suggest the ammonia produced from the agricultural industry might be manure. There are a lot of sources of ammonia in general. And ammonia is used in livestock. It is a natural process that produces ammonia. Ammonia is more like a wanted byproduct of livestock production.
Line 51: In order to.
Line 69: extra dash for “emission”, please also fix it throughout the paper.
Line 119: please use the correct degree sign.
I would recommend the author read through the paper word for word. There are a lot of formatting issues and typos throughout the paper. There are some sentences that are misleading and confusing in the Introduction Section. I also recommend the author shorten the conclusion to just focus on the main scope of the paper.
Author Response
Answers to Editor and Reviewers
We sincerely thank the respected Editor and Reviewers for especially useful expertise, their time and helpful comments and assistance in improving the scientific article.
Comments and Suggestions for Authors
Reviewer 1
This paper focuses on the evaluation of ammonia emissions with different levels of moisture content.
Line 39: I would suggest the ammonia produced from the agricultural industry might be manure. There are a lot of sources of ammonia in general. And ammonia is used in livestock. It is a natural process that produces ammonia. Ammonia is more like a wanted byproduct of livestock production.
According to Reviewer comment, sentences were corrected for clearer understanding.
“Ammonia consists more than 90% of total NH3 in livestock and evaporates at all stages of manure formation, storage and spreading on the soil [2]. Ammonia emission processes in animal husbandry receiving more attention than other gases [3]. Numerous studies have been performed analyzing microclimatic factors, their importance and influence in cowsheds: air velocity [4,5], air temperature [6-9], relative humidity [10], manure-contaminated surface areas [11].”
According to Reviewer comment, sentences were corrected.
Line 51: In order to.
According to Reviewer comment, sentence were corrected to “In order to…”
Line 69: extra dash for “emission”, please also fix it throughout the paper.
According to Reviewer comment, words were corrected to “emissions” without dash.
Line 119: please use the correct degree sign.
According to Reviewer comment, degree sign was corrected to oC.
Comments on the Quality of English Language
I would recommend the author read through the paper word for word. There are a lot of formatting issues and typos throughout the paper. There are some sentences that are misleading and confusing in the Introduction Section.
Considering to the correct Reviewer comments, the article has been thoroughly corrected and improved. English language verified too for more clear understanding essential idea.
Because our national language is not English, we have given the hand to review and correct English to professional editors.
I also recommend the author shorten the conclusion to just focus on the main scope of the paper.
According to Reviewer comment, conclusion corrected to shorten.
„Conclusions
- Ammonia evaporation is related to the intensity of moisture evaporation from manure, a strong correlation between these parameters has been established. Ammonia evaporates up to 3.9 times more intensively from liquid manure than from solid manure. When manure is stored, water evaporates intensively from it, the humidity of the manure surface decreases and a crust forms, which reduces ammonia emission. In 50 h, the intensity of water evaporation from fresh liquid manure decreases by about 2 times, ammonia emission decreases by more than 3 times. Due to the drying of manure and the formation of a crust on the surface, the biggest changes in ammonia emission occur from liquid manure. Ammonia emission from it decreases to 230 mg h-1 m-2 intensity in 24 hours (2.0-2.3 times), from thick manure to 54 mg h-1 m-2 intensity (1.9-2, 1 time).
-Manure drying and crust formation are highly dependent on microclimate factors: air temperature, humidity, speed of movement. As air temperature, ventilation intensity (air flow over manure) changes, the factors influencing the ammonia emission process also change: manure temperature, ammonia concentration gradient above manure, manure surface moisture, and also the conditions for crust formation.
-The best results by applying complex measures to reduce air pollution in cowsheds:
ï€ increase the air humidity in the barn, if it does not exceed the recommended limits (in production cowsheds, it was established that due to high air humidity, the possibilities of applying this measure are low);
ï€ reduce the moisture content of manure by using bedding;
ï€ not breaking up the crust formed on the surface of the manure;
ï€ prevent urine from accumulating in the paths in the barn and on the manure surface;
ï€ reduce surfaces contaminated with manure.“
Comments and Suggestions for Authors
Reviewer 2
Thanks to the authors for providing this study.
This study provides practical information on: “Ammonia emissions from cattle manure under variable moisture exchange between the manure and the environment”.
Good and useful study. However, please answer the following comments:
In Abstract:
A complete reformulation of the abstract, due to overlapping information and not clearly arranged, as it is not possible to distinguish between what was done in this study and the reference studies. Also, there is a repetition between the first lines and the last lines in the abstract.
Please rephrase the abstract as follows: a simple presentation of two lines indicating the importance of the study, the treatments that were carried out in a simple and clear manner, the most important results reached in this study, and then the most important recommendations.
According to Reviewer comment, Abstract was corrected for clearer understanding to shorten, all repetition was deleted.
Abstract: When reducing ammonia emissions from cowsheds, it is recommended to reduce the ventilation intensity, air temperature in the barn, manure moisture by using bedding, manure contaminated surfaces and prevent urine from accumulating in the airways. Using the mass flow method in the wind tunnel, after research of 7 types of cattle manure of different moisture content, it was found that ammonia evaporates up to 3.9 times more intensively from liquid manure than from solid manure. There is a strong correlation between ammonia and water evaporation from manure. Ammonia emission from liquid manure decreases 2.0-2.3 times, from solid manure - 1.9-2.1 times. Different cowsheds have different opportunities to reduce air pollution and conditions for manure to dry and crusts to form on the surface. The best results will be achieved by applying complex measures to reduce air pollution.
In Introduction:
- Lines: 30-34 Appropriate references should be cited.
According to Reviewer comment, Introduction was fulfilled with appropriate reference, which was cited:
- Rules for Technological Design of Cattle Buildings ŽŪ TPT 01: 2009, approved by the Minister of Agriculture of the Republic of Lithuania in 2009. August 21 by order no. 3D – 602.
- Important: There is a contradiction between what is mentioned in lines 77-79 and what is stated in lines 102-104: this contradiction must be clarified and explained because the study is based on that.
According to Reviewer comment, Introduction was corrected.
A strong correlation between ambient air temperature and ammonia emissions is found in the temperature range from 5 to 14 oC. As the ambient temperature rises, crust formation on the manure surface intensifies. Due to the crust formed on the manure surface, the intensity of ammonia emission from manure decreases [23].
As the temperature of the manure increases, the emission of ammonia increases and the crust does not form on the surface of the manure, because the high temperature of the manure increases the formation of aqueous ammonia, as a result of which the dissociation constant and the concentration of protons in the solution increase. High manure temperatures increase the formation of gaseous ammonia and reduce the solubility of ammonia in water.
- The authors did not mention the effect of the type and conditions of animal feeding on the percentage of ammonia in the produced manure (add a paragraph about it).
It was added paragraph of information in discussion part of manuscript about animal feeding on ammonia in manure according to Reviewer comment.
“Nitrogen losses on the farm are related not only to proper manure management, but also to animal feeding. The composition of the animal diet affects the release of nitrogen and carbon into the environment. In cattle fed high-concentrate forages, the optimum crude protein concentration in the diet to reduce unwanted carbon and nitrogen losses is 14.5%. This is a good strategy to reduce NH3 emissions (Devant, M., Pérez, A., Riera, J., Grau, J., Fernández, B. and Prenafeta-Boldú, F.X., 2022. Effect of decreasing dietary crude protein in fattening calves on the emission of ammonia and greenhouse gases from manure stored under aerobic and anaerobic conditions. Animal, 16(3), p.100471). The most important ruminant nutritional strategies to reduce N emissions and found that reducing the crude protein concentration in the diet was one of the most effective methods by improving microbial urea utilization in the rumen (Reynolds, C.K. and Kristensen, N.B., 2008. Nitrogen recycling through the gut and the nitrogen economy of ruminants: an asynchronous symbiosis. Journal of animal science, 86 (suppl_14), pp. E293-E305). For example, feeding black soldier fly larvae is envisioned as a novel circular strategy to extract manure, reduce its environmental impact, and convert it into insect protein for use in animal feed. Using the stable isotope 15N NH3, at least 13% of pig manure NH3-N is incorporated into fly larval body mass, increasing NH3-N uptake into larval proteins, and thereby reducing NH3 release from manure (Parodi, A., Yao, Q., Gerrits, W.J., Mishyna, M., Lakemond, C.M., Oonincx, D.G. and Van Loon, J.J., 2022. Upgrading ammonia-nitrogen from manure into body proteins in black soldier fly larvae. Resources, Conservation and Recycling, 182, p.106343).”
In Materials and Methods:
- Add the type of barn (Dairy or fattening cows) the dung was taken from.
According to Reviewer comment, Materials and Methods section clarified by adding the type of barns the manure was taken from:
-Semi-deep dairy cowshed;
-Cold box dairy cowshed;
-Half-heated box dairy cowshed.
- What are the conditions for transporting and preserving manure from the barns to the analysis device/place? Because possible during transportation some ammonia was emitted and lost.
Considering the Reviewer note, more details are explained about transporting and preserving manure from the barns to the analysis laboratory. During transportation some ammonia was emitted and lost.
Samples are taken from all places and brought to the laboratory under the same conditions and in the same special bags from which the air does not get outside. In the laboratory, without opening the bags, through special mini holes in them and connected in parallel as several replicates to the analyzer hoses to methodically ensure the correctness and reliability of the data evaluation. Air samples were taken in the places where temperature and air humidity were measured (at a height of 1.5 m above the floor) with a special device Ecoma CSD 30 (in 10 l bags). Taking one air sample takes 25 minutes. The air sampling device is made of odor-neutral materials: stainless steel and PTFE. The cylinder is made of transparent PVC. The bags are made of Nalophan. The materials of the air sampling equipment comply with standard EN 13725. The concentration of ammonia in the air sampled in the bags was measured by sucking air from them into the gas analyzer GME700. Air suction lasts up to 100 s and this amount of air in the bag (10 l) is enough to accurately measure the ammonia concentration. At low ammonia concentrations (up to 30 ppm), the gas analyzer records a stable concentration value within 60-80 s.
In Results:
- Separate the results from the discussion.
According to Reviewer comment results section was separated from discussion section. All changes are marked in the text of the article.
- Re-discuss paragraph (3.1. Ammonia emissions from manure in a wind tunnel) well, with the need to cite appropriate references.
According to Reviewer comment, after revision, this chapter has been clarified and named more precisely, which reflects all the nuances discussed in it:
“3.1. Intensity of ammonia emissions, water evaporation dependences on differences between manure layer and surface temperatures.”
Based on the reviewer comment before all discussion references were moved to the Discussion section.
In Conclusion:
A complete reformulation and abbreviation of the conclusion: There is no definitive statement that shows the summary of the results of the current study. It should be rephrased as follows: Summary results and recommendations.
According to Reviewer comment, conclusion corrected to shorten.
„Conclusions and recommendations
- Ammonia evaporation is related to the intensity of moisture evaporation from manure, a strong correlation between these parameters has been established. Ammonia evaporates up to 3.9 times more intensively from liquid manure than from solid manure. When manure is stored, water evaporates intensively from it, the humidity of the manure surface decreases and a crust forms, which reduces ammonia emission. In 50 h, the intensity of water evaporation from fresh liquid manure decreases by about 2 times, ammonia emission decreases by more than 3 times. Due to the drying of manure and the formation of a crust on the surface, the biggest changes in ammonia emission occur from liquid manure. Ammonia emission from it decreases to 230 mg h-1 m-2 intensity in 24 hours (2.0-2.3 times), from thick manure to 54 mg h-1 m-2 intensity (1.9-2, 1 time).
-Manure drying and crust formation are highly dependent on microclimate factors: air temperature, humidity, speed of movement. As air temperature, ventilation intensity (air flow over manure) changes, the factors influencing the ammonia emission process also change: manure temperature, ammonia concentration gradient above manure, manure surface moisture, and also the conditions for crust formation.
-The best results by applying complex measures to reduce air pollution in cowsheds:
ï€ increase the air humidity in the barn, if it does not exceed the recommended limits (in production cowsheds, it was established that due to high air humidity, the possibilities of applying this measure are low);
ï€ reduce the moisture content of manure by using bedding;
ï€ not breaking up the crust formed on the surface of the manure;
ï€ prevent urine from accumulating in the paths in the barn and on the manure surface;
ï€ reduce surfaces contaminated with manure.“
Comments on the Quality of English Language
Moderate editing of English language
Considering to the correct Reviewer comments, the article has been thoroughly corrected and improved. English language verified too for more clear understanding essential idea.
Because our national language is not English, we have given the hand to review and correct English to professional editors.
Comments and Suggestions for Authors
Reviewer 3
The manuscript entitled “Ammonia emissions from cattle manure under variable moisture exchange between the manure and the environment” evaluated the process of ammonia gas evaporation from manure according design and manufacture a stand, and also the effects of various factors on ammonia emissions were analyzed by simulating the conditions in cowsheds. The manuscript was described very clear and detailed. It can be accept after minor revision.
The conclusions of the manuscript was complicated. It needs to be simplification.
According to Reviewer comment, conclusion corrected to shorten.
„Conclusions
- Ammonia evaporation is related to the intensity of moisture evaporation from manure, a strong correlation between these parameters has been established. Ammonia evaporates up to 3.9 times more intensively from liquid manure than from solid manure. When manure is stored, water evaporates intensively from it, the humidity of the manure surface decreases and a crust forms, which reduces ammonia emission. In 50 h, the intensity of water evaporation from fresh liquid manure decreases by about 2 times, ammonia emission decreases by more than 3 times. Due to the drying of manure and the formation of a crust on the surface, the biggest changes in ammonia emission occur from liquid manure. Ammonia emission from it decreases to 230 mg h-1 m-2 intensity in 24 hours (2.0-2.3 times), from thick manure to 54 mg h-1 m-2 intensity (1.9-2, 1 time).
-Manure drying and crust formation are highly dependent on microclimate factors: air temperature, humidity, speed of movement. As air temperature, ventilation intensity (air flow over manure) changes, the factors influencing the ammonia emission process also change: manure temperature, ammonia concentration gradient above manure, manure surface moisture, and also the conditions for crust formation.
-The best results by applying complex measures to reduce air pollution in cowsheds:
ï€ increase the air humidity in the barn, if it does not exceed the recommended limits (in production cowsheds, it was established that due to high air humidity, the possibilities of applying this measure are low);
ï€ reduce the moisture content of manure by using bedding;
ï€ not breaking up the crust formed on the surface of the manure;
ï€ prevent urine from accumulating in the paths in the barn and on the manure surface;
ï€ reduce surfaces contaminated with manure.“

Reviewer 2 Report
Thanks to the authors for providing this study
This study provides practical information on: “Ammonia emissions from cattle manure under variable moisture exchange between the manure and the environment”.
Good and useful study. However, please answer the following comments:
In Abstract:
A complete reformulation of the abstract, due to overlapping information and not clearly arranged, as it is not possible to distinguish between what was done in this study and the reference studies. Also, there is a repetition between the first lines and the last lines in the abstract.
Please rephrase the abstract as follows: a simple presentation of two lines indicating the importance of the study, the treatments that were carried out in a simple and clear manner, the most important results reached in this study, and then the most important recommendations.
In Introduction:
· Lines: 30-34 Appropriate references should be cited.
· Important: There is a contradiction between what is mentioned in lines 77-79 and what is stated in lines 102-104: this contradiction must be clarified and explained because the study is based on that.
· The authors did not mention the effect of the type and conditions of animal feeding on the percentage of ammonia in the produced manure (add a paragraph about it)
In Materials and Methods:
· Add the type of barn (Dairy or fattening cows) the dung was taken from.
· What are the conditions for transporting and preserving manure from the barns to the analysis device/place? Because possible during transportation some ammonia was emitted and lost.
In Results:
· Separate the results from the discussion.
· Re-discuss paragraph (3.1. Ammonia emissions from manure in a wind tunnel) well, with the need to cite appropriate references.
In Conclusion:
A complete reformulation and abbreviation of the conclusion: There is no definitive statement that shows the summary of the results of the current study. It should be rephrased as follows: Summary results and recommendations.
Moderate editing of English language
Author Response

(The authors gave the same response as above.)

Reviewer 3 Report
The manuscript entitled “Ammonia emissions from cattle manure under variable moisture exchange between the manure and the environment” evaluated the process of ammonia gas evaporation from manure according design and manufacture a stand, and also the effects of various factors on ammonia emissions were analyzed by simulating the conditions in cowsheds. The manuscript was described very clear and detailed. It can be accept after minor revision.
1. Conclusions of the manuscript was complicated. It needs to be simplification.
Author Response
Answers to Editor and Reviewers
We sincerely thank the respected Editor and Reviewers for especially useful expertise, their time and helpful comments and assistance in improving the scientific article.
Comments and Suggestions for Authors
Comments and Suggestions for Authors
Reviewer 3
The manuscript entitled “Ammonia emissions from cattle manure under variable moisture exchange between the manure and the environment” evaluated the process of ammonia gas evaporation from manure according design and manufacture a stand, and also the effects of various factors on ammonia emissions were analyzed by simulating the conditions in cowsheds. The manuscript was described very clear and detailed. It can be accept after minor revision.
The conclusions of the manuscript was complicated. It needs to be simplification.
According to Reviewer comment, conclusion corrected to shorten.
„Conclusions
- Ammonia evaporation is related to the intensity of moisture evaporation from manure, a strong correlation between these parameters has been established. Ammonia evaporates up to 3.9 times more intensively from liquid manure than from solid manure. When manure is stored, water evaporates intensively from it, the humidity of the manure surface decreases and a crust forms, which reduces ammonia emission. In 50 h, the intensity of water evaporation from fresh liquid manure decreases by about 2 times, ammonia emission decreases by more than 3 times. Due to the drying of manure and the formation of a crust on the surface, the biggest changes in ammonia emission occur from liquid manure. Ammonia emission from it decreases to 230 mg h-1 m-2 intensity in 24 hours (2.0-2.3 times), from thick manure to 54 mg h-1 m-2 intensity (1.9-2, 1 time).
-Manure drying and crust formation are highly dependent on microclimate factors: air temperature, humidity, speed of movement. As air temperature, ventilation intensity (air flow over manure) changes, the factors influencing the ammonia emission process also change: manure temperature, ammonia concentration gradient above manure, manure surface moisture, and also the conditions for crust formation.
-The best results by applying complex measures to reduce air pollution in cowsheds:
ï€ increase the air humidity in the barn, if it does not exceed the recommended limits (in production cowsheds, it was established that due to high air humidity, the possibilities of applying this measure are low);
ï€ reduce the moisture content of manure by using bedding;
ï€ not breaking up the crust formed on the surface of the manure;
ï€ prevent urine from accumulating in the paths in the barn and on the manure surface;
ï€ reduce surfaces contaminated with manure.“
